# Mutation-specific CAR T cells as precision therapy for IGLV3-21[R110] expressing high-risk chronic lymphocytic leukemia

Florian Märkl [1,16], Christoph Schultheiß [2,3,16], Murtaza Ali[4], Shih-Shih Chen[5], Marina Zintchenko[6], Lukas Egli[7], Juliane Mietz [7], Obinna Chijioke [7,8], Lisa Paschold [4], Sebastijan Spajic[1], Anne Holtermann[1], Janina Dörr [1], Sophia Stock[1], Andreas Zingg[2,9], Heinz Läubli [2,9], Ignazio Piseddu[1], David Anz[1], Marcus Dühren-von Minden[10], Tianjiao Zhang[4], Thomas Nerreter [11], Michael Hudecek [11], Susana Minguet [6,12,13], Nicholas Chiorazzi [5], Sebastian Kobold [1,14,15,17] ✉ & Mascha Binder [2,3,17] ✉

The concept of precision cell therapy targeting tumor-specific mutations is appealing but requires surface-exposed neoepitopes, which is a rarity in cancer. B cell receptors (BCR) of mature lymphoid malignancies are exceptional in that they harbor tumor-specific-stereotyped sequences in the form of point mutations that drive self-engagement of the BCR and autologous signaling. Here, we use a BCR light chain neoepitope defined by a characteristic point mutation (IGLV3-21[R110]) for selective targeting of a poor-risk subset of chronic lymphocytic leukemia (CLL) with chimeric antigen receptor (CAR) T cells. We develop murine and humanized CAR constructs expressed in T cells from healthy donors and CLL patients that eradicate IGLV3-21[R110] expressing cell lines and primary CLL cells, but neither cells expressing the non-pathogenic IGLV3-21[G110] light chain nor polyclonal healthy B cells. In vivo experiments confirm epitope-selective cytolysis in xenograft models in female mice using engrafted IGLV3-21[R110] expressing cell lines or primary CLL cells. We further demonstrate in two humanized mouse models lack of cytotoxicity towards human B cells. These data provide the basis for advanced approaches of resistance-preventive and biomarker-guided cellular targeting of functionally relevant lymphoma driver mutations sparing normal B cells.

Chronic lymphocytic leukemia (CLL) is a paradigmatic low-grade lymphoma in which the B cell receptor (BCR) plays a central biological role. The BCR landscape of CLL has been extensively studied both immunogenetically and functionally. These studies revealed recognition of distinct self-antigens through stereotyped complementarity-determining region 3 (CDR3) sequence motifs, oligomeric membrane organization as well as autonomous signaling through BCR-BCR interactions (e.g.,[1-5]). Some patients from stereotypic BCR subsets are poor-risk with only limited long-term

clinical benefit with established approaches including those that target the BCR pathway (e.g.,[6]).

Advanced immunotherapeutic approaches such as chimeric antigen receptor (CAR) modified T cells are under clinical investigation for patients with poor-risk CLL. CAR T cells against CD19, a component of the BCR complex, can provide significant activity in patients suffering from advanced or refractory CLL but the rate of complete remissions and long-term responses remain well behind that observed in other lymphoma types[7-13]. Suboptimal outcomes are due to CD19

antigen loss, CAR T cell loss or dysfunction, and complete eradication of the B cell lineage that causes clinically relevant immunosuppression in these studies. Along these lines, it appears that CAR T cells can in principle be effective in CLL. However, an ideal target should be tumor-specific and of high functional relevance to prevent downregulation or loss under selective pressure. Such novel precision approaches may help to achieve durable benefit eventually even resulting in cure, as seen in other indications. The discovery of a landscape of disease-specific sequence motifs in BCRs expressed by the malignant CLL clone opened new avenues for targeted cell therapy that may eventually be translated to other types of lymphomas.

Here, we provide proof-of-concept for the activity of bona-fide tumor-specific CAR T cells for high-risk patients with CLL that express the IGLV3-21[R110] BCR light chain. The IGLV3-21[R110] subset typically shows an aggressive clinical course[6]. IGLV3-21[R110] is expressed in 10–15% of unselected CLL patients, but overrepresented in treatment-requiring CLL. Functionally, the G-to-R exchange at position 110 of the IGLV3-21 light chain – along with several conserved amino acids also in the heavy chain – confers autonomous signaling capacity to the BCR by mediating self-interactions[14–21]. Since the IGLV3-21[R110] BCR is CLL-specific and represents a critical tumor driver, we reasoned that targeting this receptor would spare normal B cells and may have a low risk of epitope escape. At the same time, the lack of persistent B cell aplasia may be of advantage in terms of infection-mediated complications and preserved responses to vaccination. Here we develop IGLV3-21[R110]-targeted CAR T cells including humanized variants thereof and demonstrate in vitro and in vivo, including against primary CLL samples, very selective targeting and eradication, leaving the healthy B cell compartment untouched including the non-pathogenic IGLV3-21[G110] expressing cells. These results underpin the potential value of such precision approach and warrant clinical investigations.

## Results

### Anti-IGLV3-21[R110] CAR T cells exhibit epitope-selective tumor cell lysis in vitro

To target the CLL-specific IGLV3-21[R110] light chain mutation, we first utilized a murine IGLV3-21[R110]–specific antibody to generate a 2nd generation CAR with CD28-CD3ζ signaling domain for retroviral transduction of primary human healthy donor (HD) T cells (HD-αR110-mCAR1 T cells) (Fig. 1a, Supplementary Fig. 1a, b). For proof-of-principle experiments, we transduced Luciferase (Luc) overexpressing NALM-6 pre-B cells (NALM-6 Luc)[22] with the pMP71 retroviral vector encoding IGLV3-21[R110] to generate a surrogate target cell line with constitutive surface expression of a hybrid BCR containing the IGLV3-21[R110] light chain (NALM-6 Luc-R110). Co-culture of NALM-6 Luc-R110 cells with HD-αR110-mCAR1 T cells showed epitope-selective lysis of IGLV3-21[R110]-expressing lymphoid target cells, while control NALM-6 Luc cells were unaffected (Fig. 1b). CD19-directed CAR T cells (HD-αCD19-mCAR) equally lysed both cell lines (Fig. 1b), while co-incubation with an unrelated CAR product (HD-E3-SAR ctrl)[23,24] or untransduced primary human T cells (UTD) failed to lyse NALM-6 lines (Fig. 1b). These specific lysis patterns were paralleled by equivalent IFN-γ secretion patterns (Fig. 1c). Notably, HD-αR110-mCAR1 T cells preferentially expanded in co-culture with NALM-6 Luc-R110 cells (Supplementary Fig. 1c), indicating good functionality of HD-αR110-mCAR T cells. Importantly, HD-αR110-mCAR T cells were not activated by and did not lyse polyclonal human B cells (Supplementary Fig. 1d).

### Anti-R110-mCAR CAR T cells are efficacious in xenograft R110+ models

We next tested the efficacy of HD-αR110-mCAR1 T cells in NSG mice engrafted with NALM-6 Luc-R110 cells (Fig. 1d). Bioluminescence imaging showed substantial reduction of NALM-6 Luc-R110 outgrowth in mice treated with HD-αR110-mCAR1 T cells (Fig. 1e), accompanied by prolonged survival and disease eradication in 17% of treated mice

(Fig. 1e, f). Notably, injected CAR T cells persisted and persisted over time (Supplementary Fig. 1e, f). We next set up a second xenograft model using the same engineering strategy to generate OCI-Ly1 lymphoma cells expressing the IGLV3-21[R110] light chain (OCI-Ly1 Luc-GFP-R110). We also included CD19-directed CAR T cells (HD-αCD19-mCAR T cells) for head-to-head comparisons. As observed for the NALM-6 model, HD-αR110-mCAR1 T cells selectively lysed OCI-Ly1 Luc-GFP-R110 cells in co-culture experiments, while control OCI-Ly1 Luc-GFP cells were unaffected (Fig. 1g). HD-αCD19-mCAR T cells lysed OCI-Ly1 Luc-GFP cells independently of IGLV3-21[R110] light chain status (Fig. 1g). Mice engrafted with OCI-Ly1 Luc-GFP-R110 cells controlled and to some extent even cleared disease, when injected with HD-αR110-mCAR1 T or αCD19-mCAR T cells (Fig. 1h). Since some mice of the CD19 group developed graft-versus-host disease, tumor-specific survival could only be displayed up to day 50. Importantly, mice survived the 50 days of the experiment in both settings, indicative of comparable activity (Fig. 1i).

### Anti-IGLV3-21[R110] CAR T cells are selective for the pathogenic R110 point mutation

We further tested the selectivity of the CARs by creating surrogate target cells not only expressing the pathogenic IGLV3-21[R110] light chain, but also the IGLV3-21[G110] wild-type light chain as its non-pathogenic counterpart (Fig. 2a, Supplementary Fig. 2a). All currently FDA-approved CARs contain signaling domains derived from the costimulatory molecules CD28 or 4-1BB. 4-1BB-based CARs induce lower expression of exhaustion markers, more central memory T cell polarization and slower, but more persistent, tumor eradication than CD28-based CARs[25,26]. Thus, we decided to test the functionality of our anti-IGLV3-21[R110] CAR in the context of 4-1BB co-stimulation (Fig. 2b). The 4-1BB-based CAR (here called mCAR2) was expressed well in primary human HD T cells (Fig. 2c, d, Supplementary Fig. 2b). Co-culture of NALM-6 surrogate target cells with αR110-mCAR2 T cells showed epitope-selective lysis of IGLV3-21[R110]-expressing target cells, while cells expressing the non-pathogenic IGLV3-21[G110] light chain were not engaged (Fig. 2e). As expected, CD19-directed CAR T cells (αCD19-mCAR) lysed NALM-6 cells independent of IGLV3-21 status (Fig. 2e), while co-incubation with Mock or untransduced primary human T cells (UTD) proved ineffective to efficiently lyse any NALM-6 model (Fig. 2e). No signs of tonic CAR signaling resulting in the expression of activation markers (CD69, CD25 and CD137) were observed for any CAR T monoculture (Fig. 2f–h, Supplementary Fig. 2c, no target cells). The anti-IGLV3-21[R110] CARs efficiently and specifically transmitted activation signals upon ligand encounter as measured by the expression of the activation markers CD69, CD25 and CD137 (Fig. 2f–h).

### Humanization of the anti-IGLV3-21[R110] scFv sequences preserves functionality

Given the potential immunogenicity of xenogeneic protein components such as a murine scFv[27] we next humanized the anti-IGLV3-21[R110] scFv sequence to generate a CAR construct with potentially lower immunogenicity (Supplementary Table 1). We used a flow cytometry based affinity ranking assay and IGLV3-21[R110]-expressing TKO cells[28] to compare the concentration-dependent binding capabilities of the murine anti-IGLV3-21[R110] antibody and the humanized scFv. As shown in Supplementary Fig. 3a, humanization did not compromise the binding affinity of the purified scFv fragment. Next, we cloned the humanized scFv fragment in a 2nd generation CAR backbone with 4-1BB-CD3ζ costimulatory domains (Supplementary Fig. 3b) and lentivirally transduced this construct into human T cells for CAR T generation (HD-αR110-CAR) (Supplementary Fig. 3c, d). We also cloned the CAR construct into a minimal-size plasmid as proof-of-principle for future clinical testing. Co-delivery of this nanoplasmid with mRNA encoding the sleeping beauty transposase via electroporation yielded efficient and stable delivery of the transgene into T cells from healthy

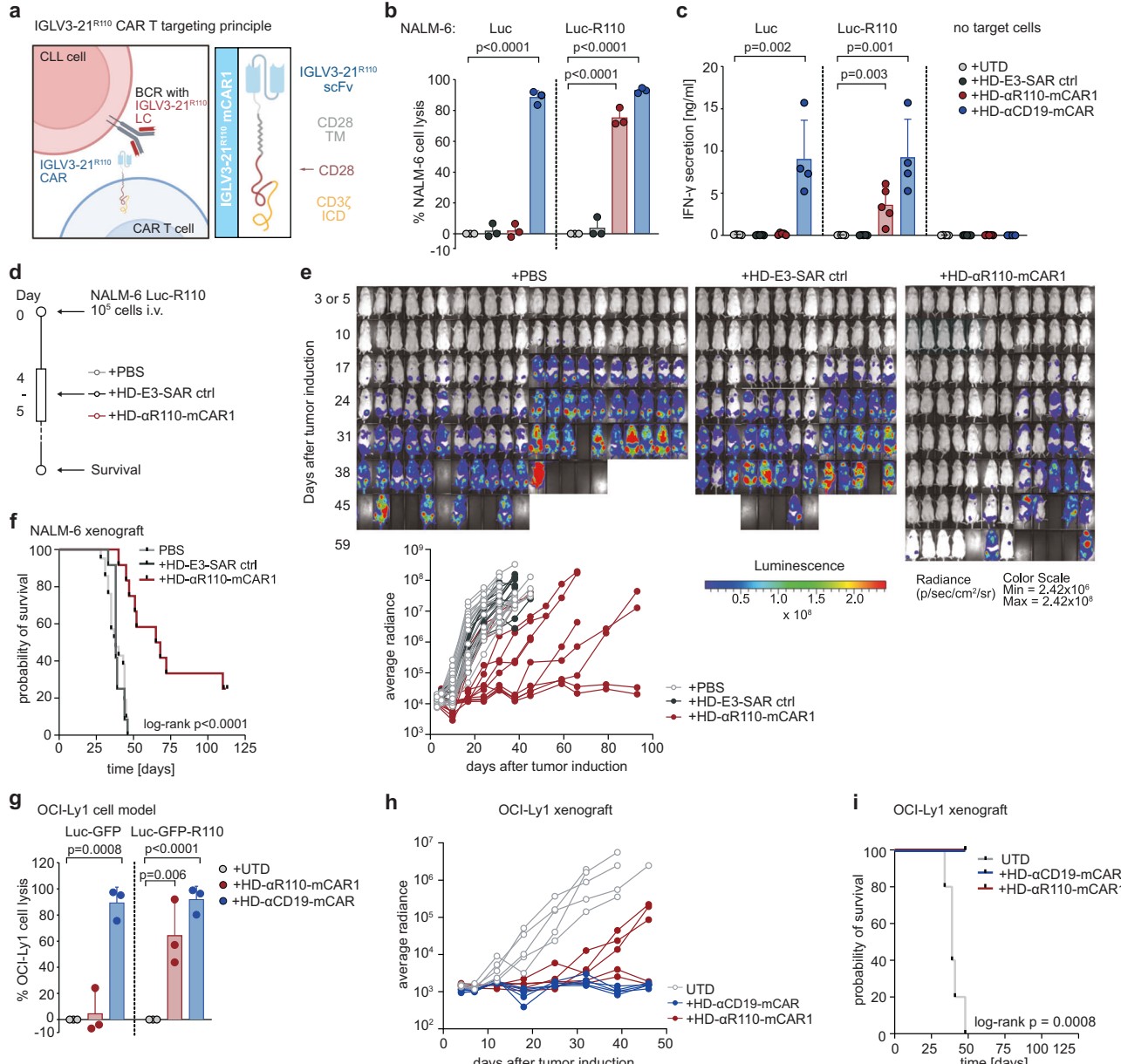

**Fig. 1 | Development of a chimeric antigen receptor (CAR) T cell targeting the IGLV3-21^R110 neoepitope. a** Schematic representation of the IGLV3-21^R110 CAR T targeting principle and the murine CAR construct (αR110-mCAR1). BCR B cell receptor, LC light chain, TMD transmembrane domain, ICD intracellular domain, Created with BioRender.com. **b** Percent tumor cell lysis based on a bioluminescence readout after 48 h co-culture of NALM-6 Luc or IGLV3-21^R110 expressing NALM-6 Luc-R110 cells with indicated CAR T cells or untransduced T cells (UTDs) in an effector to target (E:T) ratio of 0.1:1. *n* = 3 for all groups. **c** Quantification of IFN-γ secretion in cell culture supernatants after 48 h co-culture of NALM-6 Luc model. *n* = 5 in all groups but HD-αCD19-mCAR (*n* = 4). **d** Schematic representation of the workflow for the NALM-6 Luc-R110 xenograft mouse model. **e** Bioluminescence measurements of NSG mice to assess in vivo activity of HD-αR110-mCAR1 T cells from healthy donors. NALM-6 Luc-R110 growth curves are shown based on in vivo bioluminescence imaging (days 3 or 5, 10, 17, 24, 31, 38, 45, 52, 59, 65 or 66, 79, 93). NSG mice were injected i.v. with NALM-6 Luc-R110 cells and treated either four or

five days later with HD-αR110-mCAR1 T cells (*n* = 12), control HD-E3-SAR ctrl T cells (*n* = 12) or PBS vehicle solution (*n* = 21). **f** Kaplan-Meier survival plot of NALM-6 Luc-R110 mouse model. **g** Percent tumor cell lysis based on a bioluminescence readout after 48 h co-culture of OCI-Ly1 Luc-GFP or IGLV3-21^R110 expressing OCI-Ly1 Luc-GFP-R110 cells with indicated CAR T cells in an effector to target (E:T) ratio of 0.1:1. *n* = 3 in all groups. **h** Bioluminescence measurements of NSG mice engrafted with OCI-Ly1 Luc-GFP-R110 cells to assess in vivo activity of HD-αR110-mCAR1 T cells from healthy donors. OCI-Ly1 Luc-GFP-R110 growth curves are shown based on in vivo bioluminescence imaging (days 4, 7, 12, 18, 25, 32, 39, 46). NSG mice were injected i.v. with OCI-Ly1 Luc-GFP-R110 cells and treated either eight days later with HD-αR110-mCAR1 T, HD-αCD19-mCAR T, or UTD cells (for all *n* = 5). **i** Kaplan-Meier survival plot of OCI-Ly1 mouse model. All bar plots represent the indicated mean ± SD calculated from independent experiments (*n* = 3–5). Statistics: one-sided *t* test; Log-rank test for survival data. Source data are provided as a Source Data file.

individuals (Supplementary Fig. 3e, f). To test CAR T functionality in vitro, we performed co-culture killing assays with CAR T cells generated using lentiviral delivery and OCI-Ly1 cells overexpressing IGLV3-21^R110 (OCI-Ly1-R110) together with a fluorescent reporter dye indicating caspase3/7-mediated lymphoma cell apoptosis (Fig. 3a). The

obtained live cell imaging data suggested that HD-αR110-CAR T cells selectively target OCI-Ly1-R110 cells, while HD-αCD19-CAR lyse OCI-Ly1 cells independently of IGLV3-21^R110 status (Fig. 3b, c). Co-culture with an unrelated CAR targeting the human thyroid stimulating hormone receptor (TSHR) or untransduced primary T cells did not affect OCI-Ly1

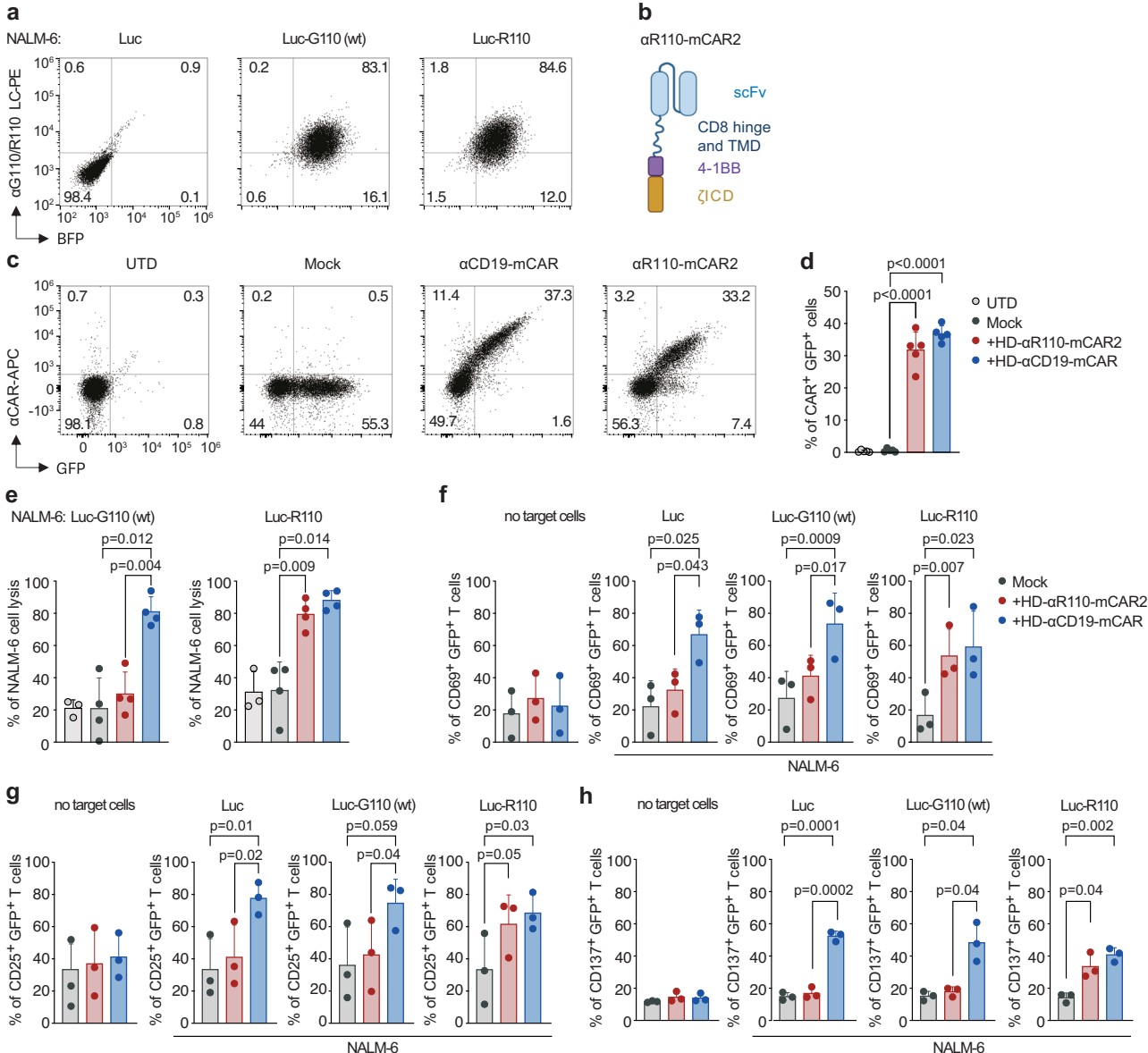

**Fig. 2 | 4-1BB-based mCAR T cells specifically targeting the IGLV3-21^R110 neoepitope. a** NALM6 pre-B cells expressing luciferase were left untreated (Luc) or transfected with a lentiviral vector encoding for the IGLV3-21^R110 light chain (LC) (NALM-6 Luc-R110) or its wild-type non-pathogenic IGLV3-21^G110 counterpart (NALM-6 Luc-G110). Representative dot plots show expression of surface BCRs containing the transfected LCs. BFP, blue fluorescent protein. **b** Schematic representation of the 4-1BB-based anti-IGLV3-21^R110 murine CAR construct (αR110-mCAR2). scFv single chain variable fragment, TMD transmembrane domain, ICD intracellular domain. Created with BioRender.com. **c** Percentage of positive CAR T cells from one representative donor and (**d**) from several donors pooled (n = 5

HDs). **e** Percent tumor cell lysis based on a bioluminescence assay after 12 h coculture of NALM-6 Luc expressing IGLV3-21^R110 or its non-pathogenic counterpart IGLV3-21^G110 in an effector to target (E:T) ratio of 5:1. (n = 4 HDs). **f–h** Expression of activation markers by CAR T cells alone or after 24 h of stimulation with the indicated target cells (1:1 ratio). (n = 3 HDs). UTDs untransduced T cells. Mock, T cells transduced with the same lentiviral vector but expressing only GFP (green fluorescent protein). All bar plots represent the indicated mean ± SD. n, indicates independent experiments/donors. Statistics: One-way ANOVA followed by t test with Welch's correction. Source data are provided as a Source Data file.

viability (Fig. 3c). Selectivity of the HD-αR110-CAR T cells was also suggested by IFN-γ-release patterns (Fig. 3d). Lastly, we expanded specificity testing of our humanized CARs. Co-culture of NALM-6 target cells with αR110-CAR T cells showed epitope-selective lysis of IGLV3-21^R110-expressing models, while cells expressing the non-pathogenic IGLV3-21^G110 wild-type light chain were only killed to background levels (Fig. 3e). CD19-directed CAR T cells lysed NALM-6 cells independent of IGLV3-21 status (Fig. 3e), while co-incubation with Mock or untransduced primary human T cells (UTD) failed to efficiently lyse any NALM-6 line (Fig. 3e).

## Healthy donor or CLL patient-derived anti-IGLV3-21^R110 CAR T cells target primary CLL cells

To come closer to the patient setting, we next asked if this targeting principle is also applicable to primary CLL cells. To identify eligible individuals for target cell isolation, we first screened a cohort of 158 CLL patients (Table 1) for IGLV3-21 status. Using light chain NGS and flow cytometry[29] we identified 17 IGLV3-21^R110 cases (Fig. 4a). Consistent with the levels of surface immunoglobulin in circulating CLL cells[5], all identified cases showed moderate but very homogeneous surface expression of the IGLV3-21^R110 neoepitope (Exemplified in Fig. 4b for

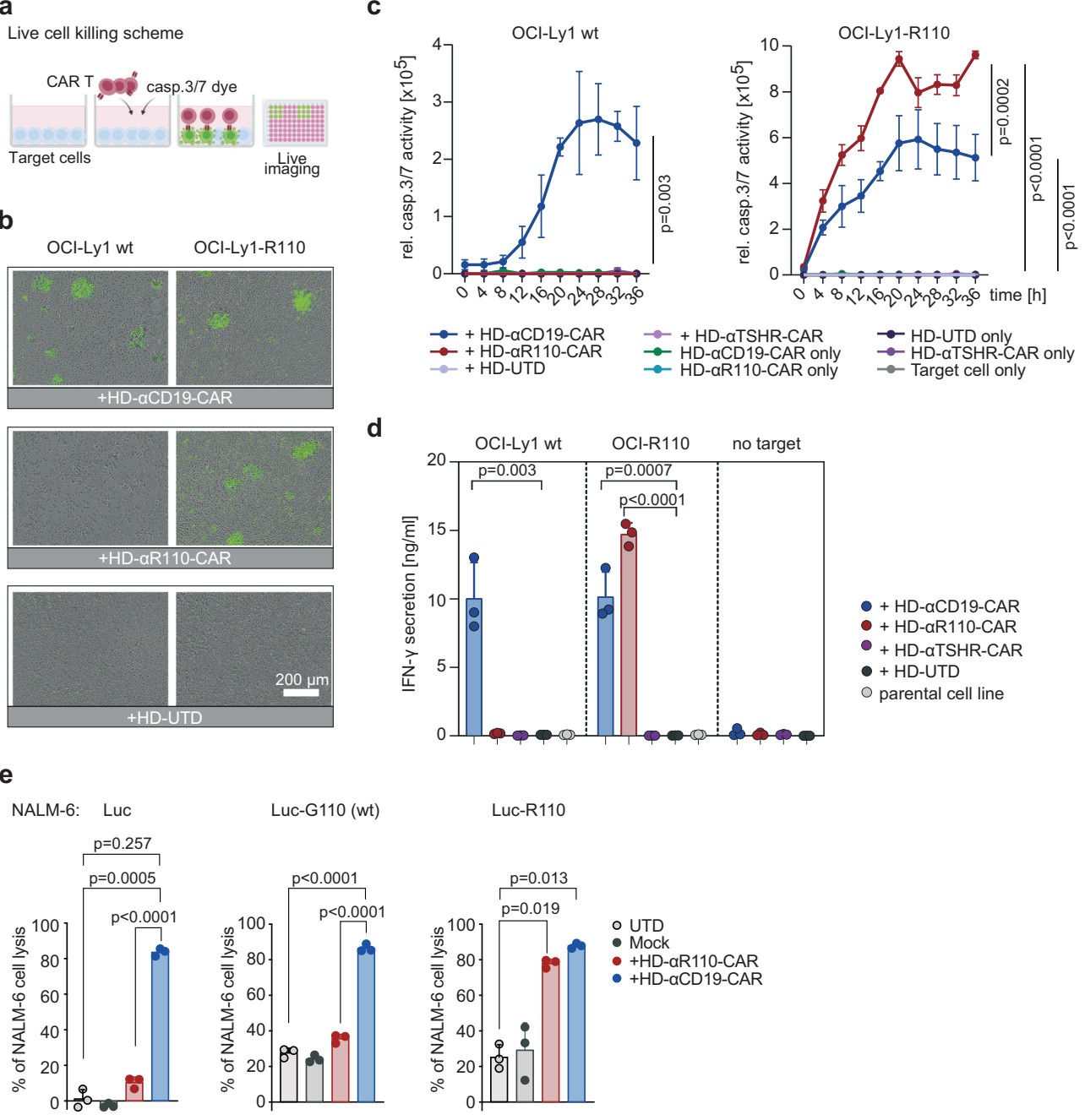

**Fig. 3 | Humanization of the binding moiety of chimeric antigen receptor (CAR) T cells against the IGLV3-21^RII0 neoepitope. a** Schematic representation of the applied live cell killing assay using the Incucyte system and the Caspase-3/7 fluorescence dye. Created with BioRender.com. **b**–**d** Cytolysis of IGLV3-21^RII0 expressing OCI-Ly1-R110 cells and OCI-Ly1 wild-type (wt) cells by healthy donor anti-IGLV3-21^RII0 CAR T cells with a humanized scFv sequence (HD-αR110-CAR) with an E:T ratio of 5:1. Cytolysis was monitored over time as caspase 3/7 activity (relative intensity of green fluorescence) in co-cultures using Incucyte S3 (**c**). Anti-thyroid stimulating hormone receptor (TSHR) CAR T cells (HD-αTSHR-CAR) and untransduced T cells (UTD) served as controls. All conditions indicated have been plotted, negative

control conditions overlap. **b** Representative images show CAR T cell-mediated cytolysis after 24 h. **d** Quantification of IFN-γ in co-culture supernatants after 24 h incubation of indicated target/effector cell combinations. **e** Percent tumor cell lysis based on a bioluminescence assay after 48 h co-culture of NALM-6 Luc expressing IGLV3-21^RII0 or its non-pathogenic counterpart IGLV3-21^GII0 in an effector to target (E:T) ratio of 5:1. All bar plots represent the indicated mean ± SD ($n = 3$) with $n$ indicates independent experiments/donors. Statistics: paired, one-sided $t$ test (**c**), one-sided $t$ test (**d**), one-way ANOVA followed by $t$ test with Welch's correction (**e**). Source data are provided as a Source Data file.

positive and negative cases; Supplementary Fig. 4a). Since staining results with the murine IGLV3-21^RII0 antibody under varying conditions of routine flow cytometry labs may differ, we set up a more standardized synthetical particle-based flow cytometry IGLV3-21^RII0 detection method. Twenty samples from the above mentioned CLL cohort were randomly selected for quantification using normalization with

synthetical beads, which showed 100% concordance with prior conventional typing results (exemplified in Fig. 4c, Supplementary Fig. 4b). Next, we selected two treatment-naïve CLL patients with or without IGLV3-21^RII0 mutation for target cell isolation. Co-culture with HD-αR110-CAR T cells showed selective lysis of IGLV3-21^RII0-positive CLL cells after 24 hours as previously demonstrated with neoepitope-

**Table 1 | Characteristics of CLL patients with IGLV3-21$^{R110}$ mutation**

| Patient | Sex | Age range | Treatment | del17q | CDR3 (aa) | IGHV | IGHD | IGHJ | IGHV status |
|---|---|---|---|---|---|---|---|---|---|
| CLL001 | m | n/a | no | no | CALDRDGMDVW | IGHV3-69-1 | IGHD3-22 | IGHJ6 | M |
| CLL010 | m | n/a | no | no | CAVDRNGMDVW | IGHV3-21 | IGHD5-18 | IGHJ6 | M |
| CLL054 | m | n/a | yes | no | CARDTHDTNGYPRWYYGLDVW | IGHV3-11 | IGHD3-22 | IGHJ6 | M |
| CLL062 | m | n/a | yes | no | CANGGGDGEYDYW | IGHV4-39 | IGHD3-10 | IGHJ4 | M |
| CLL162 | m | 50–60 | no | no | CARGVPRPHW | IGHV3-48 | IGHD1-14 | IGHJ4 | M |
| CLL306 | m | 60–70 | no | no | CARDLYYYDSSGYYSGFFDYW | IGHV1-46 | IGHD3-22 | IGHJ4 | UM |
| CLL350 | f | 70–80 | yes | no | CARDVVDYVWGSYLRAFDIW | IGHV1-69 | IGHD3-16 | IGHJ3 | UM |
| CLL362 | f | 60–70 | yes | no | CARDQVAVAGCFDYW | IGHV4-61 | IGHD6-19 | IGHJ4 | M |
| CLL374 | m | 60–70 | yes | no | CARDFVEPGYW | IGHV3-48 | IGHD6-13 | IGHJ5 | M |
| CLL381 | f | 60–70 | yes | no | CARGAGAGDYW | IGHV3-48 | IGHD6-13 | IGHJ4 | M |
| CLL385 | m | 50–60 | yes | no | CARDVGGDNSGAFDIW | IGHV2-26 | IGHD2-21 | IGHJ3 | UM |
| CLL401 | m | 70–80 | yes | no | CARDQNTMDVW | IGHV3-21 | | IGHJ6 | UM |
| CLL424 | f | 50–60 | yes | no | CARPCYDDNSDAFDIW | IGHV3-7 | IGHD3-22 | IGHJ3 | UM |
| CLL425 | f | 50–60 | yes | no | CARVENDGGYCSGGSCYPIW | IGHV3-48 | IGHD2-15 | IGHJ4 | M |
| CLL435 | m | 60–70 | no | no | CARDPGVVAATDSAIW | IGHV3-48 | IGHD2-15 | IGHJ4 | M |
| CLL438 | m | 80–90 | yes | no | CARDQNAMDVW | IGHV3-21 | | IGHJ6 | M |
| CLL442 | f | 70–80 | yes | no | CVKGGPGDGGNPFDPW | IGHV4-34 | IGHD4-23 | IGHJ5 | UM |

*f* female, *m* male, *n/a* not available, *aa* amino acid, *M* IGHV mutated, *UM* IGHV unmutated.

transduced cell lines (Fig. 4d). HD-αCD19-CAR T cells lysed CLL cells from all included patients (Fig. 4d). Co-culture with HD-αTSHR-CAR T cells or untransduced T cells had no effect on the co-culture (Fig. 4d). Cytolysis of primary CLL cells was paralleled by IFN-γ release (Fig. 4e). We then generated CAR T cells from primary T cells of two patients with CLL – one patient with active CLL (CLL433) and one in remission (CLL453) – to demonstrate their cytotoxic capacity despite the known dysfunction of T cells in this disease. Patient-derived CLL-αR110-CAR T cells showed selective lysis of OCI-Ly1-R110, while CLL-αCD19-CAR T cells from the same patients exhibited cytolysis irrespectively of the neoepitope (Fig. 4f). Next, we tested efficacy of CLL-αR110-CAR T cells derived from patient CLL425 with IGLV3-21$^{R110}$-positive active CLL in an autologous setting using a primary CLL xenograft mouse model (Fig. 4g). Application of autologous CLL-αR110-CAR T cells reduced primary CLL but not T cell load at the three week end-point in spleen and bone marrow (Fig. 4g, Supplementary Fig. 5).

## Anti-IGLV3-21$^{R110}$ CAR T cells do not mediate B cell toxicity in vitro and in vivo

Finally, we assessed the effect of HD-αR110-CAR T cells on polyclonal healthy B cells in vitro and in vivo. First, we isolated polyclonal B cells from healthy individuals and subjected them to co-culture killing assays with the different CAR T products. While polyclonal B cells were eradicated by HD-αCD19-CAR T cells, HD-αR110-CAR T cells spared this non-malignant compartment demonstrating the epitope-specificity of our targeting approach (Fig. 5a). Next, we used two humanized mouse models to show epitope selectivity of HD-αR110-CAR T cells with simultaneous sparing of healthy polyclonal human B cells in vivo (Fig. 5b, d). In the first model, human PBMCs were injected intravenously in NSG mice followed by CAR T cell injection seven days later. Quantification of blood circulating human B cells (CD19$^+$CD20$^+$) showed their persistence after eight days and subtotal eradication of B cells after 13 days when mice were treated with HD-αCD19-CAR T cells (Fig. 5c, Supplementary Fig. 6a). In contrast, treatment of mice with HD-αR110-CAR T cells did not reduce blood B cell counts after 13 days (Fig. 5c). In the second model, NFA2 mice were injected intraperitoneally with human PBMCs and either HD-αCD19-CAR or HD-αR110-CAR T cells (Fig. 5d). Quantification of human B cells in peritoneal lavage after 16 h showed subtotal reduction of B cell counts (CD19$^+$)

when NFA2 mice were treated with HD-αCD19-CAR T but not with HD-αR110-CAR T cells (Fig. 5e, Supplementary Fig. 6b).

## Discussion

CAR T cells are now a mainstay of therapy for B cell malignancies that results in long-term remission in many patients and has the potential of cure[30–35]. The potency of B cell-directed CAR T cell therapy is also highlighted by the recently reported success of treating refractory systemic lupus erythematosus[36] or refractory myasthenia gravis[37]. Since current CAR T products under development target common (co-) activation markers on the surface of B lymphocytes like CD19, CD20, CD22 and BCMA, these therapies have the drawback of eradicating the B cell lineage or substantial parts of it. As a consequence, B cell-depleted patients are more susceptible for infections complicating clinical management. Another drawback of such strategy is that most B cell-related antigens are not functionally relevant to the cancer cell. As a consequence, target expression, can get lost or mutated to prevent CAR engagement. An approach hitting a disease-driving antigen would stand lower chances of loss or downregulation. Given that target antigen modulation or loss under therapeutic pressure is a well-documented clinical issue affecting up to 50% of patients[38], the clinical significance of such an approach becomes apparent.

To respond to these challenges, we engineered a selective CAR T constructs that target a recurrent oncogenic point mutation in the BCR light chain of malignant CLL cells. We demonstrate that this approach is feasible and provide in vitro and in vivo data for the selectivity of these CAR T cells towards engineered cell lines as well as primary CLL cells from patients with the IGLV3-21$^{R110}$ mutation. The observed efficacies of the in vitro cell killing models were comparable to those observed for CD19-directed CAR T cells. Most importantly, we did not observe CAR T-mediated cytotoxicity towards healthy B cells in two humanized mouse models, arguing for the safety of our CAR T product. The specificity of this approach is demonstrated, since T cells expressing either the murine or the humanized CARs selectively killed IGLV3-21$^{R110}$ expressing target cells, while cells expressing the non-pathogenic IGLV3-21$^{G110}$ wild-type light chain were unaffected.

Notably, we generated all data on primary CLL using CARs with a humanized anti-IGLV3-21$^{R110}$ scFv that had identical binding capacities as its murine counterpart. We also did not notice any general

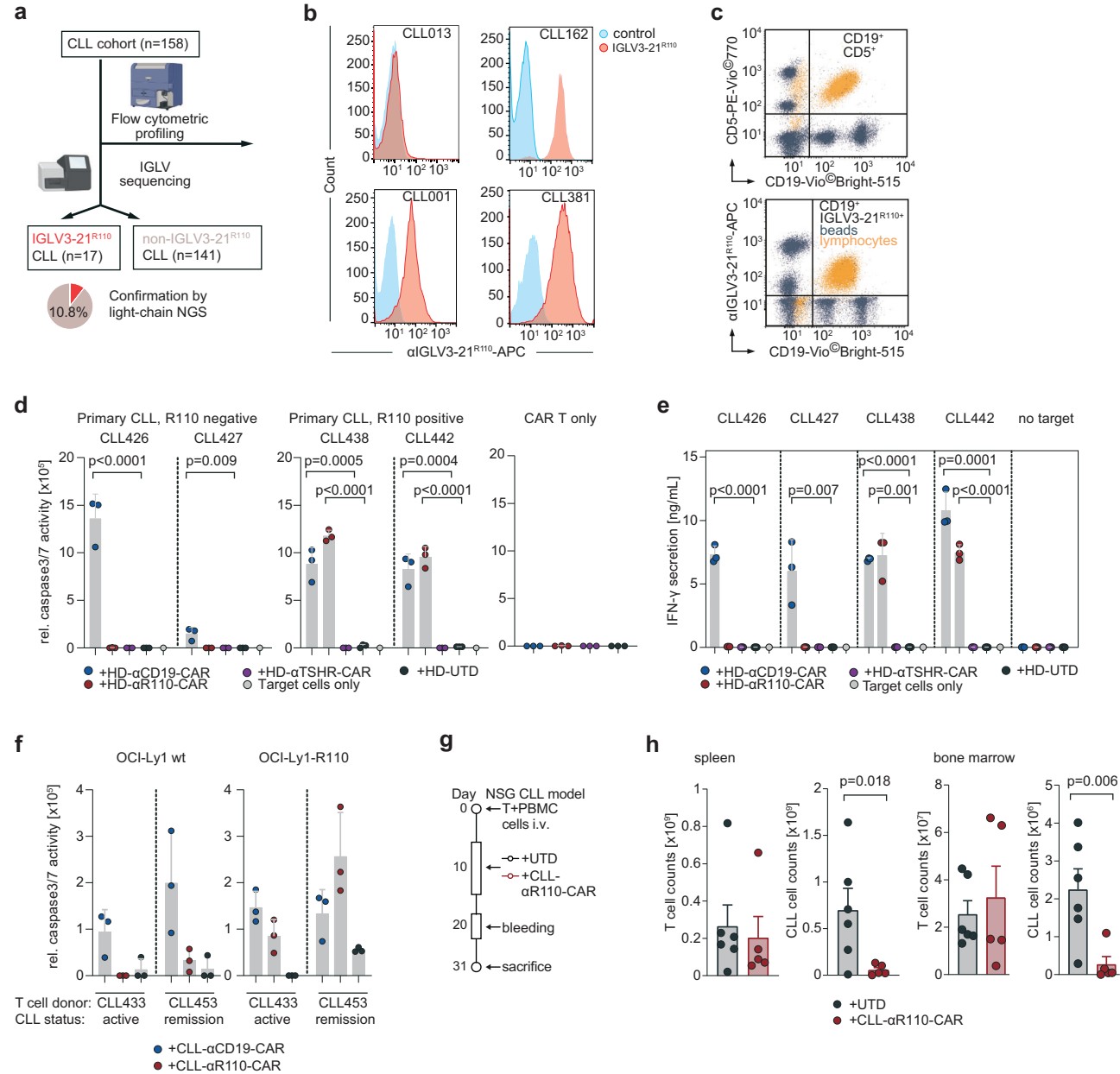

**Fig. 4 | In vitro and in vivo activity of IGLV3-21^R110 healthy donor and CLL patient-derived CAR T cells against primary chronic lymphocytic leukemia (CLL) cells. a** IGLV3-21^R110 light chain Screening work-flow of 158 CLL patients. Created with BioRender.com. **b** Exemplary results of three IGLV3-21^R110-positive and one IGLV3-21^R110-negative CLL case in a single color flow cytometric assay using APC-labelled IGLV3-21^R110-specific antibody shown as histogram. **c** Exemplary staining of one IGLV3-21^R110-positive CLL case with a bead-based assay (ApLife™ FastScreen_CLL) with triple staining of CD19, CD5 (CD19^-CD5^+ unaffected T cells) and IGLV3-21^R110. **d** Cytolysis of freshly isolated primary CLL cells from IGLV3-21^R110-positive (CLL438, CLL442) and IGLV3-21^R110-negative (CLL426 and CLL427) CLL cases by different healthy donor derived CAR T cells including HD-αR110-CAR and anti-TSHR control CAR T cells (HD-αR110-CAR) as indicated and as compared to untransduced cells (UTD). The assay was conducted as in Fig. 2b–d; the 24 h time point is shown. All groups as independent triplicates, but the target only control (n = 1). **e** Quantification of IFN-γ in co-culture supernatants after 24 h incubation of indicated target/effector cell combinations of the assay shown in panel (**d**). All

groups as independent triplicates, but the target only control (n = 1).
**f** Quantification of OCI-Ly1-R110 cytolysis mediated by CLL patient-derived CAR T cells as compared to untransduced cells (UTD). The assay was conducted with two CLL patients serving as T cell donors, one with active CLL (CLL433) and one with CLL in remission (CLL453). The assay was performed as described in Fig. 2b–d; the 24 h time point is shown. Dots in individual patients represent technical replicates (n = 3). **g** Workflow for the patient-derived xenograft mouse model for CLL. **h** NSG mice were used to assess in vivo activity of CLL-αR110-CAR T cells from CLL donor CLL425 with IGLV3-21^R110-positive CLL. Each mouse was injected i.v. with 0.5 million T cells and 20 million CLL cells collected from patient CLL425. Mice were i.p.-treated 10 days later with 7 million CLL-αR110-CAR T cells (n = 6) or untransduced cells (UTD, n = 6) from the same patient. Mice were then sacrificed at week 3 post CAR T cell injection. Only n = 5 measurements are shown for the CLL-CAR-αR110 T cell treated group since one mouse died of unknown reason. All bar plots represent the indicated mean ± SD. Statistics: one-sided *t* test. Source data are provided as a Source Data file.

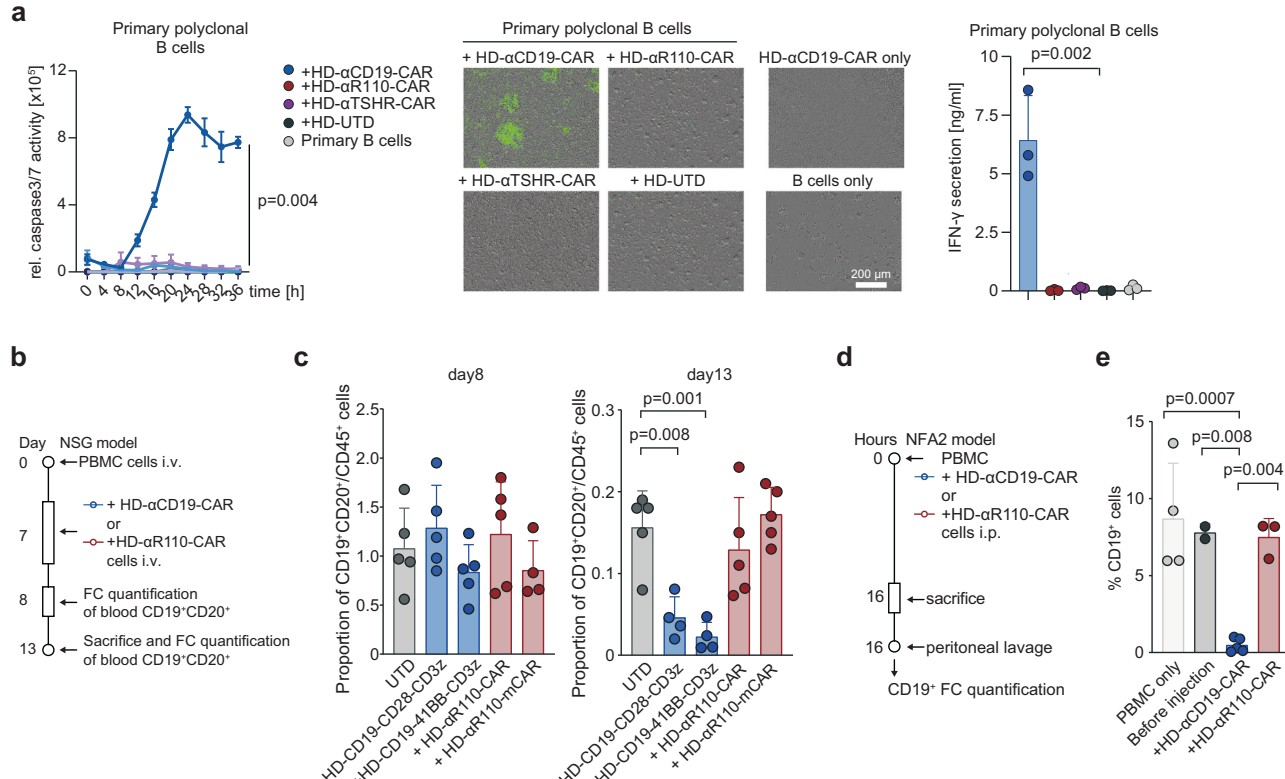

**Fig. 5 | IGLV3-21<sup>R110</sup> directed CAR T cells spare polyclonal healthy human B cells in vitro and in two humanized mouse models. a** Quantification of polyclonal B cell cytolysis mediated by healthy donor-derived CAR T cells as compared to untransduced cells (UTD) (*n* = 3 independent experiments). The assay was conducted as those in Fig. 2b–d. Representative images show CAR T cell-mediated cytolysis after 24 h. All conditions indicated have been plotted, negative control conditions overlap. **b**–**e** Humanized mouse models to test the specificity of the αR110-CAR T cells. **b** Workflow of humanized NSG model. **c** Human PBMCs were injected intravenously (i.v.) in NSG mice followed by i.v. injection of UTD, HD-

αR110-CAR T, or HD-αCD19-CAR T cells (for all *n* = 5) on day 7. The abundance of CD19⁺CD20⁺ B cells relative to all CD45⁺ cells was quantified in the blood on day 8 and 13 using flow cytometry (FC). **d** Workflow of humanized NFA2 model. **e** Human PBMCs were intraperitoneally (i.p.) injected into NFA2 mice (*n* = 4) along with HD-αR110-CAR T or HD-αCD19-CAR T cells. After 16 h mice were sacrificed, peritoneal cells were harvested by peritoneal lavage and quantified using FC. All bar plots represent the indicated mean ± SD. Statistics: one-sided *t* test. Source data are provided as a Source Data file.

disparities in cell killing capacities when using humanized CARs. Humanization of sequences during antibody or CAR T cell development represents the current state-of-the-art in the field, although most of the approved CD19-targeting CAR T cells are based on murine scFvs mainly for historical reasons (e.g. FMC63)[39,40]. These murine CD19 CARs have demonstrated great efficacy in different hematological malignancies but several reports indicate their immunogenicity including adverse clinical events like anaphylaxis[41–43]. In addition, humanization can yield CAR T products of superior function as demonstrated for FMC63[40]. Considering these points, we expect a fully humanized anti-IGLV3-21<sup>R110</sup> CAR product as the better product for initial clinical testing.

Potential clinical applications of IGLV3-21<sup>R110</sup>-targeting CAR T cells range from treatment of relapsed/refractory disease to consolidation after insufficient response to standard first-line CLL treatment. The curative potential of our approach, however, needs further evaluation in clinical trials. While the presented xenografting experiments using engineered cell models demonstrate this potential, complete tumor eradication was not achieved in all treated mice. Since overexpression of the IGLV3-21<sup>R110</sup> light chain does not confer any relevant biological function to the applied cell models, therapeutic pressure after treatment with IGLV3-21<sup>R110</sup>-targeting CAR T cells most likely favors selection of negative escape variants in this model system. This is also in line with the better performance of CD19 CAR T cells in these models. Another important confounder in xenograft studies employing human lymphocytes and tumor cells is the influence of xenoreactivity (graft

versus host reactions) in general and in our study in particular. Such effect is not a surrogate of any known side effects in clinical trials but renders data interpretation difficult, as it ranges from mild to severe symptoms, requiring abortion. Similarly, such unspecific T cell activation may also enhance T cell activity and thereby inflate therapeutic effects. As signs of GvHD were observed in Fig. 1i, we censored by day 50 to avoid interpretation bias. This constitutes a limitation to this study. Nevertheless, it is important to note that CD19-targeting CAR T cells cannot serve as reasonable comparator here given the artificial nature of the experimental setting and the fact that there is no approved CAR T product for CLL demonstrating clinical efficacy beyond individual patients in early clinical studies[33,44,45]. This is also true for the recently published primary analysis of the TRANSCEND CLL 004 study including 117 patients with relapsed or refractory CLL, which showed efficacy of CD19 CAR T cells only in a subset of patients[46]. Future studies will need to investigate whether IGLV3-21<sup>R110</sup> mutated CLL patients respond to such CD19 CAR T cells at all. In any case, the here applied and more relevant primary CLL xenograft model shows nearly complete eradication of engrafted primary CLL R110⁺ B lymphocytes cells under anti-IGLV3-21<sup>R110</sup> CAR T therapy. Notably, primary CLL mice models also come with limitations hampering survival analyzes as surrogate for long-term clinical efficacy in CLL patients. The main drawback here - besides the technically challenging requirement of concomitant engraftment and expansion of autologous T cells - is the fact, that CLL cells only transiently engraft and rather survive in a steady state than proliferate in the host[47–50]. To the

best of our knowledge any long-term therapeutic model using primary CLL cells in vivo has yet to be found.

In summary, we have developed and provided evidence of the activity and selectivity of a tumor-specific, biomarker-driven cellular targeting approach for a hematological malignancy. The crucial role of the targeted IGLV3-21[R110] neoantigen as oncogenic driver and its uniform surface expression across patients may render it an ideal target for CAR T cell therapy. Our work aligns with the endeavors of various research groups currently striving to create CAR T cells with highly specific targeting for lymphoma cells[51,52] or autoreactive B cells[53].

## Methods

All studies using primary human material complied with all relevant ethical regulations of the approving committees. The here reported human studies were approved by the ethics committees of the Universities of Hamburg–Eppendorf (PV3400, PV4767) and Halle-Wittenberg (2014-075). All studies with mice were approved by the respective board/committee and performed in accordance with the respective animal welfare regulations. These committees were the Government of Oberbayern (ROB; ROB-55.2–2532.Vet_02-20-208, ROB-55.2–2532.Vet_02-17-135 and ROB-55.2-2532.Vet_02-20-109), the Institutional Animal Care and Use Committee (IACUC) of the Feinstein Institute for Medical Research (2012-032) and the veterinary office of the canton of Zurich, Switzerland (protocol ZH049/20).

### Patient cohort

Blood samples from 158 CLL patients were collected after informed consent including approval to use the presented clinical information in Table 1 for research publication at the Universities of Hamburg–Eppendorf, Freiburg and Halle-Wittenberg. This cohort had a median age of 67 years (range 43–86) and consisted to 69% of male and 31% female participants. Sex and gender were not relevant for the findings and interpretation of this study and thus not included in the further study design. Healthy donor blood cells were obtained from anonymous volunteers as buffy coat from the blood banks of the University Hospitals Halle (Saale), Munich and Basel after informed consent. Peripheral mononuclear cells (PBMCs) were isolated by Ficoll gradient centrifugation, resuspended in FCS + 10% DMSO and cryopreserved in liquid nitrogen. IGLV3-21[R110] expressing CLL was characterized by flow cytometry and next-generation sequencing (NGS) of the light chain loci as described in[29,54–62] and the following sections.

### IGLV3-21[R110] flow cytometry

IGLV3-21[R110] expression on CLL cells was tested with an APC-/PE-labelled IGLV3-21[R110]-specific antibody or by an PE-labelled IGLV3-21[R110/G110] antibody (1:100) recognizing both light chains (AVA Lifescience GmbH, Denzlingen, Germany) using patient-derived PBMCs. Twenty cases were additionally analyzed with the ApLife™ FastScreen[CLL] assay that uses CD19, CD5 and IGLV3-21[R110] antibodies in addition to labelled spheres to define cut-off levels comparable throughout different measurements. The ApLife™ FastScreen[CLL] assay was applied according to the manufacturer's instructions.

### IGLV3-21[R110] next-generation sequencing (NGS)

IGL repertoires were profiled as described[54,57–60,62]. Sequencing libraries were generated from 250 ng genomic DNA isolated from PBMCs using the GenElute mammalian genomic DNA miniprep kit (Sigma-Aldrich, Taufkirchen, Germany, #G1N10-1KT). The IGL primer pool was adapted from[63] to cover the complete IGLJ (FR4) region including the first nucleotide of the triplet for amino acid position 110 at the junction of IGLJ and IGLC. The sequences of the new reverse primers are (5′–3′): GTGAGACAGGCTGGG, CAAGAGCGGGGAAGG, CAACTTGGCAGG-GAAAG, GGGAGACCAGGGAAG, TCACCCTAGACCCAAAAG. Sequencing was performed on an IlluminaMiSeq (paired-end, 2×301-cycles, v3 chemistry). The MiXCR framework v3.0.12[64] with the IMGT v3 IGH

library[65] as reference for sequence alignment was used for clonotype assembly. Amino acid position 110 was defined by nucleotide 28 of the FR4 region.

### Cell lines and primary CLL and healthy donor blood cells

OCI-Ly1 (ACC 722) and NALM-6 (ACC 128) were purchased from the DSMZ (German Collection of Microorganisms and Cell Cultures GmbH). Luciferase (Luc) overexpressing cell line NALM-6 Luc was previously described[22]. OCI-Ly1 Luc-GFP was generated as described[66]. For ectopic expression of the IGLV3-21[R110] light chain, the coding sequence was cloned into the Lentiviral Gene Ontology (LeGO) vector LeGO-iC2-Puro+ via AsiSI/EcoRI (for expression in OCI-Ly1), the retroviral vector pMP71 or the lentiviral vector pHRSin containing and IRES-mTAGBFP2 element for transduction efficiency analysis (for expression in NALM-6 Luc), respectively[67]. In addition, the IGLV3-21[G110] light chain coding sequence was cloned into the lentiviral vector pHRSin (for expression in NALM-6 Luc).

### CAR constructs

The CAR construct derived from the murine single-chain variable fragment (scFv) of the IGLV3-21[R110]-specific antibody from AVA Lifescience GmbH (Denzlingen, Germany). It was cloned into the retroviral vector pMP71 containing CD28 and CD3ζ costimulatory domains or the lentiviral vector pCDH containing 4-1BB and the intracellular domain of ζ followed by a T2A-copGFP element for transduction efficiency analysis. The scFv sequence derived from the murine anti-IGLV3-21[R110] antibody was humanized and cloned via NheI and RsrII into the backbone of an established lentiviral CD19 CAR vector containing 4-1BB and CD3ζ costimulatory domains[68]. This humanized anti-IGLV3-21[R110] CAR construct was also cloned into a minimal-size plasmid containing the sleeping beauty inverted terminal repeats (ITR) for transposon-based gene delivery. The Human thyrotropin receptor-directed CAR T cells (αTSHR-CAR)[69] as well as a published CD19-targeting CAR (αCD19-CAR)[68] served as control. These CARs contained a truncated epidermal growth factor receptor (tEGFR) for sorting of CAR-expressing cells[70]. All sequences are listed in Supplementary Table 1.

### Virus production

Lentivirus production was performed as described earlier[71]. For retrovirus production, 293Vec-Galv and 293Vec-RD114 cell lines[72] were used (kind gift of Manuel Caruso, Québec, Canada). Retroviral pMP71 vectors (kindly provided by C. Baum, Hannover) carrying the sequence of the relevant receptor were stably introduced in packaging cell lines[66]. Single cell clones were generated and indirectly screened for virus production by determining transduction efficiency of primary T cells. This method was used to generate the producer cell lines 293Vec-RD114 for scFv-R110-CD28-CD3ζ (αR110-mCAR), EGFRvIII-CD28-CD3ζ (E3 synthetic agonistic receptor (E3-SAR)) and scFv-CD19-CD28-CD3ζ (αCD19-mCAR (WO2015187528A1)).

### Generation of CAR-expressing primary human T cells

Pan T cells were isolated from healthy donor or CLL patient-derived whole blood (Pan T Cell Isolation Kit and Auto MACS Quant, Miltenyi, Bergisch Gladbach, Germany), stimulated with CD3/CD28 T-cell activation Dynabeads (Life Technologies, Carlsbad, USA, #11131D) at a 1:1 bead to cell ratio, and lentivirally transduced 24 h later at a multiplicity of infection of 1.5 or retrovirally transduced 48 h after isolation[73]. Alternatively, peripheral blood mononuclear cells (PBMCs) were isolated by density centrifugation (Pancoll). T cells were activated with anti-CD3, anti-CD28 antibodies (1 µg/ml) and 500 U/ml of human IL-2, and lentivirally transduced 48 h later at a multiplicity of infection of 4. All T cells were expanded in complete T cell medium supplemented with penicillin–streptomycin (100 U/mL; Life Technologies, Carlsbad, USA)) and fed IL-2 (50 U/mL; Stem Cell Technologies, Vancouver,

Canada, #78036 or 100 U/mL, Peprotech, Cranbury, USA, #200-02) every 48 hours. Dynabeads were removed day 6 after isolation. For generation of transposon-based CAR T cells, Pan T cells were isolated and stimulated with CTS CD3/CD28 Dynabeads (Thermo Fisher #40203D) in PRIME-XV T Cell CDM medium (Irvine Scientific #91154) supplemented with 10% heat-inactivated human serum and 600 IU/ml IL-2 for three days. For electroporation, activated T cells (100–200 million/ml) were electroporated with 150 µg/ml transposon-plasmid DNA and 30 µg/ml transposase-encoding mRNA (eTheRNA) using the Maxcyte electroporator with the "Expanded T-cell 3" protocol. Beads were removed 72 h after electroporation. Efficiency of gene delivery was determined using flow cytometry and antibodies directed against EGFR (Cetuximab, Merck KGaA, 1:1000) and the G4S linker in the scFv (clone E7O2V, Cell Signaling; #63670, 1:200). Cetuximab was detected with the anti-human IgG-FITC secondary antibody (Sigma Aldrich #F4512, 1:1000), the G4S linker with the anti-Rabbit IgG H&L-PE (Invitrogen, #P-2771MP, 1:1000).

### Antibody and scFv affinity ranking

The affinity of the murine anti-IGLV3-21[RII0] antibody from AVA Lifescience and the humanized single-chain variable fragment (scFv) was determined using flow cytometry and the TKO cell model[28]. For ectopic expression of the IGLV3-21[RII0] light chain in TKO cells, the coding sequence was cloned into the vector pMIZYN. Transduction was performed after generation of lentiviral particles in 293 T cells as described above. For binding, $2 \times 10^5$ TKO cells were seeded as duplicates in 96-well plates and incubated with serial dilutions of antibody/scFv for 30 min at 4 °C followed by secondary detection (anti-human-IgG1-APC, Clone IS11-12E4.23.20, Miltenyi, Bergisch Gladbach, Germany, #130-119-944, 1:50) and quantification on a MACSQuant Analyzer 10 flow cytometer (Miltenyi, Bergisch Gladbach, Germany).

### In vitro cytotoxicity assay and cytokine quantification

For Incucyte S3 assays, target cells seeded at $2 \times 10^4$ cells/well in a 96-well plate were co-incubated with effector cells at effector-to-target (E:T) ratio 5:1 in complete media. Primary CLL target cells were isolated by Ficoll gradient centrifugation. Polyclonal control B cells from healthy donors were isolated by Dynabeads™ CD19 isolation kit from Invitrogen (#11351D) after Ficoll gradient centrifugation. CAR T cell-mediated tumor cell cytotoxicity was assessed using the Incucyte Caspase-3/7 Reagent (BioScience, Essen, Germany, #4440). Other cytotoxicity assays were performed using a flow cytometry-based readout (BD LSRFortessa (BD Biosciences, New Jersey, USA)) after 48 h of coculture with human CAR T cells. Dead cells were stained using the violet fixable viability dye (BioLegend, San Diego, USA, #423113) for 10 min at room temperature. Following this, cell surface proteins were stained for 20 min at 4 °C. Tumor cells were quantified by using an anti-CD19-BV785 antibody (clone 6D5) (BioLegend, San Diego, USA, #115543, 1:200). For the quantification of the CAR T cells, the CD3-APC (clone OKT3) (#317318), CD4-PerCP/Cy5.5 (clone OKT4) (#317428), CD8a-BV605 (clone RPA-T8) (#301040), EGFR-AlexaFluor488 (clone AY13) (#352901) (all from BioLegend, San Diego, USA, 1:200) and c-Myc-FITC (SH1-26E7.1.3, Miltenyi Biotech, Bergisch-Gladbach, Germany, #130-116-485, 1:50) antibodies were used. Furthermore, luciferase-based toxicity assays were performed using Bio-Glo Luciferase Assay System (#G7940, Promega Corporation, USA). Alternatively, luciferase-based cytotoxicity was performed as described[74]. In addition, cytokine measurements were done by ELISA (BD Biosciences, Frankllin Lakes, USA, #555142) or a bead-based immunoassay technology (LEGENDplex B cell panel #740526, Biolegend, San Diego, USA). Values below the limit of detection were considered zero.

### Activation assay and flow cytometry analysis

For analysis of extracellular activation markers, cells were incubated at a ratio of 1:1 with NALM-6 for 24 h, collected and stained for 15 min at 4 °C in the dark. Acquisition was performed in the Attune NxT Acoustic Focusing Cytometer (Invitrogen). The antibodies used were: Biotin Strep-Tag (NWSHPQFEK) (Genscript, clone 5A9F9, #A01736, 1:2000), APC Streptavidin (BioLegend #405207, 1:200), PE-anti-hCD25 (BioLegend, clone BC96, #302606, 1:200), PE-Cy7-anti-hCD137 (Invitrogen, clone 4B4-1, #24-1379-42, 1:50), APC-anti-hCD69 (Invitrogen clone CH/4, #MHCD6905, 1:200) and BV421-anti hCD3 (BioLegend, clone UCHT1, #300434, 1:200). Analysis was performed using the FlowJo Software v.10.

### CAR T cell in vivo assays

Experiments were performed with female NSG (NOD.Cg-Prkdc[scid] Il2rg[tm1Wjl]/SzJ) mice aged 2–3 months from Janvier Labs ($n = 85$) according to the regulations of the ROB. The mice were kept in facilities with a 12-hour dark-light cycle, including a 30-minute twilight period. The humidity in the facilities was between 45 and 60% and the average temperature was between 20 and 22 °C. Sex was not relevant for this xenograft models and was thus not considered in the study design. NALM-6 and OCI-Ly1 Luc-R110 xenograft models were established in NSG mice following the intravenous (i.v.) injection of $10^5$ tumor cells in 100 µL PBS. Given the nature of the models, there was no tumor size or burden limit, as such could not be directly inferred from external measures. As tumor cells were carrying luciferase we had set radiance $\geq 10^{10}$ (photons/s/cm²/sr) as upper surrogate limit, which however was never reached in this study.

Animals were randomized into treatment groups according to tumor burden. Experiments were performed by a scientist blinded to treatment allocation and with adequate controls. No time points or mice were excluded from the experiments presented in the study. For adoptive cell therapy (ACT) studies, $4-5 \times 10^6$ active CAR T cells were injected i.v. in 100 µl PBS. Persistence of injected CAR T cells was monitored by flow cytometric quantification of transduced T cells (CD3 + EGFR) in 100 µl blood. For T cell tracking in murine blood the CD3-APC (clone OKT3) (#317318), EGFR-AlexaFluor488 (clone AY13) (#352901) (all from BioLegend, San Diego, USA, 1:200), c-Myc-FITC (SH1-26E7.1.3, Miltenyi Biotech, Bergisch-Gladbach, Germany, #130-116-485, 1:50), CD8-BV605 (clone SK1, BioLegend, #344742), CD4-PerCP-Cy5.5 (clone OKT4, BioLegend, #317428) and CD19-BV785 (clone 6D5, BioLegend, #115543, 1:200) were used. Tumor burden was measured using a luciferase-based IVIS Lumina X5 imaging system (PerkinElmer, Waltham, USA). Parameters such as weight loss >15%, body conditioning score, hunched posture, paralysis or behavioral changes were used as humane surrogate endpoints and are further defined as survival. Survival analyses were recorded in Kaplan Meyer plots.

### Patient-derived CLL xenograft

T cells derived from PBMCs of patient CLL425 were activated with CD3/CD28 dynabeads (Life Technologies, Carlsbad, USA, #11131D) and IL-2 (Company, Article#, concentration) for 3 days. The activated T cells were then mixed with CLL425 PBMCs at the ratio of 1 to 40. Total $0.5 \times 10^6$ T cells and $20 \times 10^6$ PBMCs were intravenously injected into each NSG (NOD/SCID/IL2rγnull) mouse, and a total of 12 mice were injected. 10 days after, mice were randomly split into two groups, 6 mice were intraperitoneally injected with 7 million CLL-αR110-CAR T cells ($n = 6$), or untransduced T cells (UTD) generated from the same patient. Three weeks later, all the mice were sacrificed, blood, spleen and bone marrow samples were collected for the number of T and CLL cells. All animal experiments were approved by the Institutional Animal Care and Use Committee (IACUC) of The Feinstein Institute of Medical Research. All procedures were performed in accordance with guidelines of IACUC. Age- and sex-matched NSG mice (8 weeks old) were housed to adapt to a cycle of a 12-hour-light/dark (lights on/off at 7 am/7 pm) at least 2 weeks before the start of experimentation. The room was maintained at $23 \pm 2$ °C and at a constant humidity. All mice

were housed in cages with filter tops and fed food ad libitum. Sex was not relevant for this xenograft models and was thus not considered in the study design. Single cell suspension prepared from these tissues were then stained and analyzed by flow cytometry for the number of human hCD45+CD19+CD5+ CLL B cells and hCD45+CD19-CD5+ T cells. For flow cytometry the anti-mCD45-PE-Cy7 (clone 30-F11, BioLegend, #103114, 1:200), anti-hCD45-AlexaFluor700 (clone HI30, BioLegend, #304024, 1:200), anti-CD19-PE (clone HIB19, BioLegend, #9 82402, 1:100) and anti-CD5-AlexaFluor488 (clone UCHT2, BioLegend, #300632, 1:200) antibodies were used.

## Humanized mouse models

Human PBMCs were isolated by gradient centrifugation. Female NSG mice were purchased from Janvier Labs (in total: $n = 25$) and injected at 6–8 weeks i.v. with $20 \times 10^6$ PBMCs. On day seven the mice were injected i.v. with $1.5 \times 10^6$ active αR110-CAR T or αCD19-CAR T cells. The frequency of CD19+ CD20+ cells of all human CD45+ cells was analyzed by flow cytometry (d8,13) using the CD45 PerCP-Cy5.5 (clone 2D1, BioLegend, #368504), anti-CD19-BV785 (clone 6D5, BioLegend, #115543), anti-CD20-APC (clone 2H7, BioLegend, #302309), anti-CD4-BV605 (clone OKT4, BioLegend, #317438) and anti-CD8 BV421 (clone SK1, BioLegend, #344748) antibodies. In a separate assay, human PBMCs were labelled with the fluorescent membrane marker PKH26 (Sigma-Aldrich, MINI26-1KT). For the in vivo killing assay, 3–5 month old NFA2 (NOD.Cg-*Rag1*$^{tm1Mom}$ *Flt3*$^{tm1Irl}$ *Mcph1*$^{Tg(HLA-A2.1)1Enge}$ *Il2rg*$^{tm1Wjl}$/J) mice were injected i.p. with $2.5 \times 10^6$ PKH26-labelled PBMC and $1.25 \times 10^6$ αR110-CAR T or αCD19-CAR T cells (in total $n = 18$). NFA2 mice were housed under a 12 h light/12 h dark cycle (lights on: 6 am, lights off: 6 pm) at temperatures from 21–24 °C with 35–70% humidity. Mice were sacrificed after 16 h, peritoneal cells were harvested by peritoneal lavage and quantified using flow cytometry. Mice in which either PBMC or CAR T cells could not be detected by flow cytometry were excluded from the dataset to control for unsuccessful i.p. injections. The following markers were used: anti-human (h)CD45-Pacific Blue (2D1, Biolegend, San Diego, USA, #368540, 1:200), anti-mouse (m)CD45-APC (30-F11, Biolegend, #103111, 1:100), anti-human CD19-BV785 (HIB19, Biolegend, #302240, 1:100), anti-human CD3-FITC (UCHT1, Biolegend, #300406, 1:200), anti-human CD14-Cy5.5 (TuK4, Invitrogen, Carlsbad, USA, #MHCD1418, 1:500), anti-human CD56-PE-Cy7 (NCAM16.2, BD Franklin Lakes, USA, #335791, 1:200), Zombie Aqua live/dead stain (Biolegend, #423102, 1:500).

## Statistical analysis

One-tailed student's *t*-test was used for comparisons between two groups. One way ANOVA was used when more than two groups were compared. A log-rank (Mantel-Cox) test was used to compare survival curves. All statistical tests were performed with GraphPad Prism software (v8.3.0). No statistical methods were used to predetermine sample size.

## Reporting summary

Further information on research design is available in the Nature Portfolio Reporting Summary linked to this article.

# Data availability

NGS data is deposited at the European Nucleotide Archive (ENA) under the accession number PRJEB65274. The remaining data are available within the Article, Supplementary Information or Source Data file. Source data are provided with this paper.

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

## Acknowledgements

The authors thank Christoph Wosiek, Aline Patzschke, Yiqing Du and Katrin Nerger for excellent technical assistance. Financial support from the Deutsche Forschungsgemeinschaft (DFG BI 1711/4-1 to M.B.) and intramural Roux program funding (to M.B.) is acknowledged. We acknowledge the iFlow Core Facility of the LMU University Hospital and Alexander Navarrete-Santos of the Cell Sorting Core Facility of the Martin-Luther-University (MLU) Halle-Wittenberg as well as the flow cytometry core facility at the Department of Biomedicine at the University Hospital Basel and Jelena Markovic-Djuric, Mihaela Barbu-Stevanovic, Cécile Cumin and Morgane Hilpert for assistance with the generation of flow cytometry data. We also thank Nadine Bley and the Core Facility Imaging of the MLU for help with live cell imaging. This study was further supported by the International Doctoral Program iTarget: Immunotargeting of Cancer funded by the Elite Network of Bavaria (S.K.), the Melanoma Research Alliance Grants 409510 (to S.K.), the Marie-Sklodowska-Curie Program Training Network for Optimizing Adoptive T Cell Therapy of Cancer funded by the H2020 Program of the European Union (Grant 955575, to S.K.), the German Cancer Aid (AvantCAR.de to S.K.), the Else-Kröner-Fresenius-Stiftung (IOLIN to S.K.), the Wilhelm-Sander-Stiftung (to S.K.), the Deutsche José Carreras Leukämie-Stiftung (to S.K.), the Monika-Kutzner-Stiftung (to S.K.), the Fritz-Bender-Stiftung (to S.K.) LMU Munich's Institutional Strategy LMUexcellent within the framework of the German Excellence Initiative (to S.K.), the Bundesministerium für Bildung und Forschung Projects Oncoattract, CONTRACT and Beyondantibody (S.K.), by the Bavarian Research Foundation (BAYCELLATOR to S.K.), by the European Research Council Grant 756017, 101100460 and 101124203 (to S.K.), Deutsche Forschungsgemeinschaft (DFG; KO5055-2-1 and 510821390 to S.K.), the Bruno und Helene Jöster Foundation (to S.K.) and by the SFB- TRR 338/1 (2021–452881907 to S.K.). S.M. is supported by DFG through BIOSS - EXC294 and CIBSS - EXC 2189, SFB1479 (Project ID: 441891347 - P15) MI 1942/4-1 (Project ID: 501418856) and MI1942/5-1 (Project ID: 501436442 to S.M.). M.Z. is supported by FOR2799 (MI1942/3-1 to S.M.).

## Author contributions

Idea & design of research project: S.K., M.B., S.M.; Supply of critical material (e.g. patient material, mouse models, cohorts): M.B., N.C., M.D.M., T.N., M.H., O.C., L.E., J.M.; Establishment of Methods: S.K., M.B., F.M., M.A., C.S., L.P., S.-S.C., M.Z., N.C., T.N., M.H., M.D.M., A.Z., H.L.; Experimental work: F.M., M.A., C.S., S.-S.C., M.Z., L.P., T.Z., SeS., A.H., J.D., SoS., A.Z.; Analysis and interpretation of primary data: M.B., S.K., C.S., F.M., M.A., L.P., S.-S.C., N.C., M.Z., I.P., D.A., S.M.; Drafting of manuscript: M.B., S.K., C.S. All authors reviewed and revised the manuscript.

## Competing interests

M.D.M. discloses to be shareholder of AVA-Lifescience GmbH and inventor of patent application WO/2023/152204 (ANTIBODIES TARGETING THE B-CELL RECEPTOR OF CHRONIC LYMPHOCYTIC LEUKEMIA AND USES THEREOF). The use of the afore mentioned antibody sequences in CAR T cells is subject to the current patent application EP22186810.2 by M.D.M. and M.B. S.K., M.H. and T.N. are inventors of several patents in the field of cellular therapies. S.K. has received honoraria from BMS, GSK, Galapagos, Cymab, Novartis, Miltenyi Biomedicines and TCR2 Inc. S.K. has received license payments from TCR2 Inc and Carina Biotech. S.K. received research support from Arcus Biosciences, Plectonic GmbH, Tabby Therapeutics, Catalym GmbH and TCR2 Inc for work unrelated to this manuscript. All other authors disclose no potential conflicts of interest.

## Additional information

[1]Division of Clinical Pharmacology, Klinikum der Universität München, Munich, Germany. [2]Division of Medical Oncology, University Hospital Basel, Basel, Switzerland. [3]Laboratory of Translational Immuno-Oncology, Department of Biomedicine, University and University Hospital Basel, Basel, Switzerland. [4]Internal Medicine IV, Oncology/Hematology, Martin-Luther-University Halle-Wittenberg, Halle (Saale), Germany. [5]Karches Center for Oncology Research,

The Feinstein Institutes for Medical Research, Northwell Health, Manhasset, NY, USA. [6]Faculty of Biology, University of Freiburg, Freiburg, Germany. [7]Cellular Immunotherapy, Institute of Experimental Immunology, University of Zurich, Zurich, Switzerland. [8]Institute of Pathology and Medical Genetics, University Hospital Basel, Basel, Switzerland. [9]Laboratory of Cancer Immunotherapy, Department of Biomedicine, University and University Hospital Basel, Basel, Switzerland. [10]AVA-lifescience GmbH, Ferdinand-Porsche-Straße 5/1, Denzlingen, Germany. [11]Medizinische Klinik und Poliklinik II, Universitätsklinikum Würzburg, Würzburg, Germany. [12]Signalling Research Centres BIOSS and CIBSS, University of Freiburg, Freiburg, Germany. [13]Center of Chronic Immuno-deficiency CCI, University Clinics and Medical Faculty, Freiburg, Germany. [14]German Cancer Consortium (DKTK), Partner Site Munich, Munich, Germany. [15]Einheit für Klinische Pharmakologie (EKLiP), Helmholtz Munich, Research Center for Environmental Health (HMGU), Neuherberg, Germany. [16]These authors contributed equally: Florian Märkl, Christoph Schultheiß. [17]These authors jointly supervised this work: Sebastian Kobold, Mascha Binder.
✉e-mail: sebastian.kobold@med.uni-muenchen.de; Mascha.Binder@unibas.ch

