## [Peer Review File · Nature Communications]

Mutation-specific CAR T cells as precision therapy for IGLV3-21R110 expressing high-risk chronic lymphocytic leukemiaEditorial Note: This manuscript has been previously reviewed at another journal that is not operating a transparent peer review scheme. This document only contains reviewer comments and rebuttal letters for versions considered at *Nature Communications*.

REVIEWERS' COMMENTS

Reviewer #1 (Remarks to the Author):

In this further revised manuscript, the authors expanded on their discussion, and added new data for in-vivo expansion of CART cells.

Strengths of this manuscript include the novelty as no BCR targeted car T-cell therapy has been reported to date; the manuscript is overall well written; and the data are presented in a systematic way. However, there are major issues, as outlined below:

- Efficacy of the CAR: this reviewer is concerned that the antitumor efficacy of the CAR is generally low to modest, thus limiting the translational potential for this therapy. Figure 1E indicates that only 2/10 mice had durable response following treatment with R110 CART cells.

- Controls: as this manuscript aims to develop a new therapeutic option for B-cell malignancies, comparisons to CART19 as the gold standard should be included and tested in different B-cell models (including Nalm6, as requested by more than 1 reviewer). These data should also be included in the main manuscript figures rather than the supplementary figures. I appreciate the authors' point of view that this therapy is aimed to be developed for CLL and agree that this is an area of unmet needs where no FDA approved CART cell therapy is available yet (although the TRANSCEND-004 clinical trial was recently reported to be positive for liso-cel in CLL and the data are currently under review with FDA, Siddiqi et al, Lancet 2023). However, developing a suboptimal CAR in CLL would be more concerning, since the efficacy of CART19 in CLL is inferior to start with, compared to its efficacy in other B-cell malignancies.

- Appreciate the new discussion with regard to the rationale for using humanized clone.

Since the rationale for CAR humanization is not specifically hypothesis driven, I recommend moving the data to a supplementary figure.

REVIEWERS' COMMENTS

Reviewer #1 (Remarks to the Author):

In this further revised manuscript, the authors expanded on their discussion, and added new data for in-vivo expansion of CART cells.

Strengths of this manuscript include the novelty as no BCR targeted car T-cell therapy has been reported to date; the manuscript is overall well written; and the data are presented in a systematic way. However, there are major issues, as outlined below:

We thank the reviewer for the fruitful discussion throughout the whole publication process and also really appreciate all comments to advance our manuscript. We have addressed the remaining points as follows.

- Efficacy of the CAR: this reviewer is concerned that the antitumor efficacy of the CAR is generally low to modest, thus limiting the translational potential for this therapy. Figure 1E indicates that only 2/10 mice had durable response following treatment with R110 CART cells.

As discussed in the previous point by point letters, we concur with the notion that higher response rates are always more desirable for patient benefit and have previously expanded our data discussion in this direction. We however respectfully disagree with the interpretation on the translational relevance of the artificial overexpression systems utilized in this study, where the R110-mutated BCR is not a driver of neither the NALM6 nor the OCI-LY1 model. The purpose was to demonstrate efficacy against control transduced T cells in aggressive and fast growing models in a randomized and blinded fashion. Both studies have met their primary endpoint, thus confirming efficacy with the previously discussed limitation that R110-BCR is not relevant to these models. As a consequence, we already observed in vitro that lysis was never total (Figure 1) and discuss this limitation accordingly. When we transitioned to the more relevant patient-derived model, using primary CLL cells in vitro and in vivo similar total or subtotal CLL elimination, supporting further investigation and translation of the strategy (Figure 4), which is currently being prepared at our institution.

- Controls: as this manuscript aims to develop a new therapeutic option for B-cell malignancies, comparisons to CART19 as the gold standard should be included and tested in different B-cell models (including Nalm6, as requested by more than 1 reviewer). These data should also be included in the main manuscript figures rather than the supplementary figures. I appreciate the authors' point of view that this therapy is aimed to be developed for CLL and agree that this is an area of unmet needs where no FDA approved CART cell therapy is available yet (although the TRANSCEND-004 clinical trial was recently reported to be positive for liso-cel in CLL and the data are currently under review with FDA, Siddiqi et al, Lancet 2023). However, developing a suboptimal CAR in CLL would be more concerning, since the efficacy of CART19 in CLL is inferior to start with, compared to its efficacy in other B-cell malignancies.

We would like to reiterate the argument, that we do not aim to develop a new therapeutic option for B cell malignancies, but rather provide proof-of-principle that a distinct subgroup of CLL patients with aggressive disease course can be targeted with a CAR T product that specifically recognizes a surface-expressed driver mutation on malignant but not healthy cells. We understand that CD19 CARs – given their efficacy in other B cell malignancies or autoimmunity and their widespread use to experimentally validate the functionality of novel CAR backbones – are considered as important comparator in the community. For this reason, we re-included all available CD19 CAR T data in the main figures.

Nevertheless, we also want to highlight that in the context of the present specific research question addressed in this manuscript, CD19 CAR T cells cannot serve as the efficacy benchmark given their (not yet) proven efficacy in CLL patients. We thank the reviewer for mentioning the first data from TRANSCEND trial. While this trial reports CR in 18/92 patients and significant efficacy only in the group with previous BTK and venetoclax failure, there is no general benefit of CD19 CAR T treatment. It remains to be seen if the regulators will see this as sufficient for approval or label extension but if so will need to be carefully scrutinized for impact on the R110-mutated patient population. We included this point in the discussion.

- Appreciate the new discussion with regard to the rationale for using humanized clone. Since the rationale for CAR humanization is not specifically hypothesis driven, I recommend moving the data to a supplementary figure.

Agreed. The humanization approach is now presented in Supplementary Figure 3a, b.